# Optimization Methods as a Base for Decision Making in Land Consolidation Projects Ranking

**Goran Marinković** [1], **Zoran Ilić** [1], **Milan Trifković** [2], **Jelena Tatalović** [2,*] and **Marko Božić** [3]

1   Faculty of Technical Sciences, University of Novi Sad, 21000 Novi Sad, Serbia
2   Faculty of Civil Engineering Subotica, University of Novi Sad, 24000 Subotica, Serbia
3   Meixner d.o.o., Hermanova 16/G, 10000 Zagreb, Croatia
*   Correspondence: lazicjelena@uns.ac.rs; Tel.: +381-642-735-082

**Abstract:** Land consolidation (LC) is an activity that brings numerous benefits to rural areas. However, being resource demanding, the LC requires a decision on where it should be provided or where the limited resources should be distributed in order to maximize its effects. In order to avoid the subjective decision maker's preferences, optimization methods for identifying the priorities are recommended. Bearing in mind that every optimization method could give different results, we proposed the utilization of multiple optimization methods for ranking the cadastral municipalities which are candidates for providing LC. In this research, the main aim was to find if it is possible to avoid the subjective decision making in cadastral municipalities (CM) as a candidate for LC ranking by utilizing the statistical approach. Additionally, in this research, the analysis was provided, varying the number of optimization criteria. In this research, two assumptions were adopted: (1) every single optimization method has the same weight, and (2) the differences between different ranks are results of random errors. After determining the average ranking of a certain cadastral municipality, its interval of ranking is calculated by using the Student's distribution. Cadastral municipalities that belong within the interval of available resources are candidates for providing LC. In the case study, fifteen cadastral municipalities were researched, including eight and ten criteria for optimization, and results showed that there are significant differences between ranks of cadastral municipalities varying depending on the method utilized.

**Keywords:** integral assessment; risk reducing; rural area; cadastral municipality; standard deviation; null hypothesis

## 1. Introduction

Land consolidation (LC) is recognized as a process that is important and effective for sustainable development and spatial planning of rural areas but is also expensive and long-lasting [1].

The importance of land consolidation and its positive effects on agricultural land effective use and rural revitalization are thoroughly researched. Even though the positive effects of land consolidation have been well known for centuries, it is still not provided on a satisfactory level, and their positive effects are not exhausted yet. According to Sayilan [2], even though the LC works in Turkey were initiated in 1961, sufficient success could not be achieved. According to research by Yin et al. [3], based on 92 peer-reviewed articles, "rural land consolidation had played more positive than no/negative roles in facilitating rural revitalization, with cases reporting generally positive outcomes accounting for 74%." The main rural issues recognized, such as land fragmentation, eco-environmental destruction, industrial lag and rural hollowing, could be alleviated by proper utilization of LC [4]. The paper by Ying et al. [5] stated that "scant attention has been paid to the relationship between population hollowing and rural sustainability", and one conclusion stated that the difference of rural hollowing in different villages was significant and that per capita of the

village was one of three variables which affect the household directly. Land fragmentation has numerous negative effects on agricultural production, which cause a decrease in yield, an increase in treatment duration, fuel cost and time required to access the parcels, and consequently decreases in agricultural profitability [6]. Bearing in mind these results, it is obvious that LC could play a significant role in the alleviation of rural hollowing, also containing significant possibilities for rural revitalization.

Starting from the assumption that the LC process is long-lasting and expensive, it could be stated that it is not possible to provide enough financial and other necessary resources (experts, technical and logistic) in order to realize all required LC processes at the same time and on rural areas which need revitalization. This fact causes priority determination and choosing appropriate models for proper decision making. In order to avoid the subjective decision, different optimization methods are utilized in the issue of prioritization of land consolidation [7–11].

Since the complexity of land consolidation caused by numerous possible criteria and limited resources, as well as the imperfections of optimization methods, it is questionable if there exists an optimization method that could result in the best solution. The optimization method for decision making could be based on the different approaches and could have similarities and differences [12], consequently resulting in different ranks of alternatives. In order to avoid the uncertainties in different optimization methods, it is possible to utilize a combination of them to find the optimal solution. Tomić et al. [13] utilized PROMETHEE I, PROMETHEE II, TOPSIS, WSM-EW and WSM-AHP methods in order to calculate the land consolidation suitability index. Concluding that "the ranking results was more dependent on indicator weights than the choice of multi criteria analysis (MCA)", they recommend the utilization of the simplest WSM method and that the decision-makers review and approve indicators' weighting (preferences). This conclusion immediately implicates that the ranking is significantly dependent on the decision-makers' ability to choose the proper weighting for each utilized method.

In this research, we start from the assumption that chosen weights are burdened with errors and that it is not possible to determine their true values and that, consequently, resulted ranking is also burdened by random errors. This characteristic of optimization methods allows the statistical analysis of obtained results, and it is the main issue of this research. The rank analysis was provided by introducing 8 and 10 criteria, and obtained results were compared.

The analysis of CM ranking in this research is based on finding the average ranking of each municipality and the standard deviation of ranks. The average ranking and the standard deviation define the span in which the rank could belong with the same probability. In the case of limited resources, it is possible to determine the number of CM which could be consolidated. The candidates for land consolidation are ranked better or equal to the number of land consolidations defined in advance.

## 2. Materials and Methods

The research area covers the territory of the municipality of Bela Crkva, located in the South Banat district of the Autonomous Province of Vojvodina in the Republic of Serbia. The research included the areas unavailable for construction development of fifteen cadastral municipalities with a total area of 33,630 hectares, 68,094 parcels and 10,968 future land consolidation participants. The material used for quantifying the criterion of each alternative was obtained from the database of the Real Estate Cadastre (SKN) of Bela Crkva, the Secretariat for Agriculture of the Municipality of Bela Crkva and the Statistical Office of the Republic of Serbia.

The main criteria originated from the main goals and tasks of LC are ranked from $f_1$ to $f_8$. The additional criteria $f_9$ and $f_{10}$ were introduced in order to research their influence on cadastral municipalities (CM) ranking related to the main goals of LC. This research was provided firstly by all 10 criteria and after that with the 8 basic criteria.

This database is, in fact, the basis of the Republic Geodetic Institute for the Municipality of Bela Crkva, and it is located in the Bela Crkva Real Estate Cadastre Department (RCD). This database contains information about all properties on the territory of the municipality. From it are taken the data on the areas unavailable for construction development (Table 1):

- (f1) The share of agricultural land in the total area of the land consolidation area;
- (f2) Average surface area of cadastral parcels for each land consolidation area;
- (f3) Average number of cadastral parcels per participant for each land consolidation area;
- (f4) Average surface area of the participants' property for each land consolidation area.

**Table 1.** Data taken from the database of the RCD of Bela Crkva.

| CM_Name | CM_ID | $f_1$ (%) | $f_2$ (ha/Parcels) | $f_3$ (Parcels) | $f_4$ (ha) |
|---|---|---|---|---|---|
| Ban. Palanka 1 | 800317 | 25.54 | 0.49 | 10.12 | 4.93 |
| Ban. Subotica | 800660 | 75.82 | 0.56 | 6.09 | 3.39 |
| Bela Crkva | 800333 | 74.39 | 1.30 | 3.22 | 4.19 |
| Vračev Gaj 1 | 800643 | 60.71 | 0.43 | 5.47 | 2.35 |
| Vračev Gaj 2 | 800732 | 77.03 | 0.60 | 3.18 | 1.91 |
| Grebenac | 800449 | 57.25 | 0.46 | 5.44 | 2.53 |
| Dobričevo | 800678 | 84.04 | 0.92 | 7.02 | 6.48 |
| Dupljaja | 800392 | 53.70 | 0.41 | 7.21 | 2.94 |
| Jasenovo | 800465 | 79.43 | 0.43 | 5.51 | 2.39 |
| Kajtasovo | 800473 | 11.76 | 0.80 | 4.44 | 3.57 |
| Kaluđerovo 1 | 800481 | 68.56 | 0.47 | 13.86 | 6.24 |
| Kruščica | 800503 | 87.27 | 0.36 | 9.43 | 3.43 |
| Kusići 1 i 2 | 800708 | 76.39 | 0.32 | 6.17 | 2.01 |
| Kusići 3 | 800724 | 64.79 | 0.42 | 17.17 | 7.28 |
| Crvena Crkva | 800341 | 78.94 | 0.50 | 5.61 | 2.78 |

This database is the local base of the Municipality of Bela Crkva, and it contains data on agricultural land and estates. The data on the future land consolidation projects are taken from this base, i.e., (Table 2):

- (f5) Percentage of farmers owning property larger than 5 ha;
- (f6) Share of the state property out of the total agricultural land.

**Table 2.** Data taken from the database of the Secretariat for Agriculture of the Municipality of Bela Crkva.

| CM_ID | $f_5$ (%) | $f_6$ (%) |
|---|---|---|
| 800317 | 10.99 | 53.5 |
| 800660 | 16.42 | 22.0 |
| 800333 | 2.44 | 29.7 |
| 800643 | 9.71 | 25.2 |
| 800732 | 5.41 | 38.2 |
| 800449 | 10.97 | 23.2 |
| 800678 | 16.35 | 48.0 |
| 800392 | 13.89 | 24.1 |
| 800465 | 10.88 | 15.7 |
| 800473 | 2.88 | 75.2 |
| 800481 | 24.52 | 23.9 |
| 800503 | 13.48 | 9.5 |
| 800708 | 8.71 | 16.1 |
| 800724 | 5.13 | 59.9 |
| 800341 | 8.23 | 19.2 |

The Statistical Office of the Republic of Serbia (RZS) is a special professional organization in the state administration system in the Republic of Serbia. In the system of official statistics of the Republic of Serbia, according to the provisions of the Law on Official Statistics, the RZS performs activities within the system of the Republic of Serbia as

the principal participant. The RZS, among other things, carries out activities related to economic production, the implementation of the census, conducting household research, carrying out research relating to the economy and agriculture, and the introduction and management of statistical registers, with the exception of certain financial sector research.

From this database, the data on active agricultural population (criterion $f_7$) in the future land consolidation areas are collected (Table 3).

**Table 3.** Active agricultural population by land consolidation areas.

| CM_ID | Working Population | Active Agricultural Population | Share ($f_7$) [%] |
|---|---|---|---|
| 800317 | 497 | 370 | 74.4 |
| 800660 | 109 | 82 | 75.2 |
| 800333 | 7005 | 396 | 5.7 |
| 800643 | 1028 | 528 | 51.4 |
| 800732 | 1028 | 528 | 51.4 |
| 800449 | 582 | 520 | 89.3 |
| 800678 | 157 | 140 | 89.2 |
| 800392 | 515 | 418 | 81.2 |
| 800465 | 898 | 330 | 36.7 |
| 800473 | 175 | 156 | 89.1 |
| 800481 | 72 | 58 | 80.6 |
| 800503 | 627 | 347 | 55.3 |
| 800708 | 864 | 389 | 45.0 |
| 800724 | 864 | 389 | 45.0 |
| 800341 | 471 | 134 | 28.5 |

The total costs of land consolidation and cost per hectare (criterion $f_8$) are explicated in EUR and given in Table 4.

**Table 4.** Review of the total costs of realization of land consolidation projects.

| CM_ID | Consolidated Surface Area [ha] | Cost [EUR] | $f_8$ [EUR/ha] |
|---|---|---|---|
| 800317 | 3230 | 458,763 | 142 |
| 800660 | 910 | 144,817 | 159 |
| 800333 | 3436 | 458,692 | 133 |
| 800643 | 2856 | 458,342 | 160 |
| 800732 | 283 | 64,767 | 229 |
| 800449 | 3848 | 624,613 | 162 |
| 800678 | 1704 | 225,138 | 132 |
| 800392 | 2434 | 417,621 | 172 |
| 800465 | 2970 | 395,963 | 133 |
| 800473 | 3724 | 787,254 | 211 |
| 800481 | 967 | 167,429 | 173 |
| 800503 | 2568 | 342,563 | 133 |
| 800708 | 2342 | 327,552 | 140 |
| 800724 | 568 | 116,363 | 205 |
| 800341 | 1790 | 252,638 | 141 |

Additional criteria: (1) total annual effect and return periods (criterion $f_9$) and (2) predicted profitability of agricultural production caused by LC are given in Tables 5 and 6, respectively.

The integral assessment method is based on the utilization of 6 multi-criteria analysis methods (AHP, VIKOR, COPRAS, ELECTRE, TOPSIS and SAW) and analysis of the differences in obtained rankings for each method. Analytic Hierarchy Process (AHP) is based on the structure of hierarchy in decision problems [14]. The hierarchy consists of three levels: on the top level is the goal of the decision; at the second level, the criteria are situated and followed by alternatives on the third level. According to the same literature [14]: "Hierarchical decomposition of complex systems appears to a basic device used by the

human mind to cope with diversity." Another explication of the AHP reads, "The Analytic Hierarchy Process (AHP) is a theory of measurement through pairwise comparisons and relies on the judgements of experts to derive priority scales" [15]. AHP method is applied in different land consolidation projects expressing consent for solving problems in the land consolidation reallocation process (where the satisfaction of participants was 91.5% with the solution based on the AHP model compared with 62.7% satisfaction with the solution based on classical interview-based model) [16] and regional land consolidation [17]. Although the AHP method is developing still criticism exists, the possibility of the appearance of the Rank Reversal phenomenon, variability of set criteria as well as its construction in accordance with reasons and simplicity led to the statement that "almost all applications of AHP are potentially flawed" [18].

**Table 5.** Review of the total annual financial effect and return periods of assets invested into land consolidation ($f_9$).

| CM_ID | Total Annual Financial Effect of Land Consolidation C [EUR] | $f_9$ [Year] |
|---|---|---|
| 800317 | 47,685 | 27.0 |
| 800660 | 39,882 | 10.2 |
| 800333 | 147,737 | 8.7 |
| 800643 | 100,225 | 12.8 |
| 800732 | 12,600 | 14.4 |
| 800449 | 127,333 | 13.7 |
| 800678 | 82,770 | 7.6 |
| 800392 | 75,545 | 15.5 |
| 800465 | 136,350 | 8.1 |
| 800473 | 25,316 | 87.1 |
| 800481 | 38,321 | 12.2 |
| 800503 | 129,530 | 7.4 |
| 800708 | 103,404 | 9.1 |
| 800724 | 21,270 | 15.3 |
| 800341 | 81,671 | 8.7 |

**Table 6.** Profitability of agricultural production due to land consolidation in the adopted period.

| CM_ID | Adopted Number Years $n$ | $f_{10}$ [Un. N] |
|---|---|---|
| 800317 | 10 | 0.37 |
| 800660 | 10 | 0.98 |
| 800333 | 10 | 1.15 |
| 800643 | 10 | 0.78 |
| 800732 | 10 | 0.69 |
| 800449 | 10 | 0.73 |
| 800678 | 10 | 1.31 |
| 800392 | 10 | 0.65 |
| 800465 | 10 | 1.23 |
| 800473 | 10 | 0.11 |
| 800481 | 10 | 0.82 |
| 800503 | 10 | 1.35 |
| 800708 | 10 | 1.10 |
| 800724 | 10 | 0.65 |
| 800341 | 10 | 1.15 |

Similarities and differences between VIKOR (VIKOR is the acronym in the Serbian language which means "Multicriteria Optimization and Compromise Solution") and TOPSIS (Technique for Order of Preference by Similarity to Ideal Solution) multiple criteria decision-making methods are based on an aggregating function, which represents "closeness to the ideal", where in the VIKOR method the linear normalization, while in the TOPSIS method vector normalization is used to eliminate the units of criterion function [12]. The modi-

fied VIKOR method was utilized in solving a complex problem in the land consolidation process which included "Production dimension", "Environmental dimension" and "Life dimension" [19]. The correctness of the TOPSIS method is also examined for the possibility of increasing the efficiency of identification of land for consolidation, and results showed that "the lowest conformity of results was achieved for a combination of Hellwig's method and TOPSIS" [1]. This result indicates that combinations of methods should be chosen very carefully and proven by other methods.

The essential principle of the COPRAS (Complex PRoportional ASsessment of alternatives) method is the possibility of combining values of all indicators, and it uses classical normalization and assumes direct and proportional dependence of priority and utility degree of studying alternatives [11]. The COPRAS method and its modifications are utilized in different kinds of problems [20], including land consolidation [21].

The SAW (Simple Additive Weighting) is the oldest and most widely known practically used method [22]. The SAW method clearly shows the main concept of multi-criteria evaluation methods integrating criteria values and weights into a single magnitude. The SAW or WSM (Weighted Sum Model) could be applied effectively in many different situations during land consolidation project realization where a lot of alternatives could appear (the exchange of land parcels between participants, public facility allocation, land use suitability analysis, etc.) [23]. The multi-criteria decision analysis method ELECTRE (ELimination Et Choice Translating REality) "has a good potential to solve multi-objective problems when compensation among criteria is not allowed" [24]. The ELECTRE method consists of eight steps [24]: normalization of the decision matrix, multiplying the columns of the decision matrix by the associated weights, determining the concordance and discordance set, calculating the concordance matrix, calculating the discordance matrix, determining the concordance dominance matrix, determining the discordance dominance matrix, determining the aggregate dominance matrix and eliminating the less favorable alternatives.

The development of multi-criteria analysis through the development of multi-criteria decision-making methods showed is twofold: on the one side, it broadens the models involved, and on the other side, it broadens the mathematical models involved. This development is caused by the fact that existing methods (1) are not perfect and (2) should be improved in order to cover the different domains of practice. For example, the AHP methods combined with fuzzy sets results in FAHP [25], while the modified VIKOR method combined Fermatean hesitant fuzzy sets resulted in novel multi-attribute decision-making approach [26]. In the paper by Ozdagoglu et al. [27], the PIPRECIA [28] method is integrated with COPRAS for solving transportation problems. In the paper by Zahid et al. [29], the ELECTRE method is combined with fuzzy set theory in order to "empower it to encompass the abstinence part of vague information in addition to satisfaction and dissatisfaction in two dimensional scenarios". Deng and Chen [30] modified the TOPSIS method and combined it with intuitionistic fuzzy set (IFS) in order to describe the uncertainty better, while Zhang and Dai [31] introduced novel TOPSIS method decision-theoretic rough fuzzy sets. All of those multiple-criteria decision-making improvements result with additional expert knowledge needed in order to utilize them properly. In this paper, we propose a simple approach: to use existing models with respect to their imperfections which is explicated by using statistical methods for analysis.

The general summary of the given methods is that they are applicable to land consolidation problems, but no one could be used as an ultimate solution, i.e., every single method is at risk of shortcoming influence on final results.

In this research, we introduce the assumption that every obtained rank by every utilized method is burdened by random error, and it could be explicated in the following way:

$$R_i = R_{ij} + \epsilon_{ij} \tag{1}$$

where

-     $R_i$–right (ideal) rank of *i*-th cadastral municipality

- $R_{ij}$–rank of $i$-th cadastral municipality obtained by $j$-th method and
- $\epsilon_{ij}$–random error in ranking of $i$-th cadastral municipality by $j$-th method

In this case, the "ideal" rank of each cadastral municipality could be determined (estimated) as a mean value:

$$\overline{R_i} = \frac{1}{n} \sum_{j=1}^{n} R_{ij} \tag{2}$$

where $\overline{R_i}$ denotes the mean value of ranks for $i$-th cadastral municipality and $n$ the number of utilized methods.

The standard deviation of ranks is calculated as follows:

$$\sigma_{R_i} = \sqrt{\frac{1}{n-1} \sum_{j=1}^{n} \left(R_{ij} - \overline{R_i}\right)^2} \tag{3}$$

The interval in which could belong the rank of $i$-th cadastral municipality than is determined as follows:

$$R_i \in \left(\overline{R_i} - t * \sigma_{R_i}, \ \overline{R_i} + t * \sigma_{R_i}\right) = I \tag{4}$$

where $t = t_{f, 1-\alpha}$ is the quantile of Student's distribution, $f$ degrees of freedom and $\alpha$ is level of significance. In this research we adopt $f = 5$ and $\alpha = 0.05$. Consequently the value for the quantile of the Student's distribution is $t_{5,0.95} = 2.5076$.

The final interval of ranking $I$ is determined as follows:

$$I = \begin{cases} 1, & \left(\overline{R_i} - t * \sigma_{R_i}\right) < 1 \\ \left(\overline{R_i} - t * \sigma_{R_i}, \ \overline{R_i} + t * \sigma_{R_i}\right), & \left(\overline{R_i} - t * \sigma_{R_i}\right) \geq 1 \wedge \left(\overline{R_i} + t * \sigma_{R_i}\right) \leq n \\ n, & \left(\overline{R_i} + t * \sigma_{R_i}\right) \geq n \end{cases} \tag{5}$$

## 3. Results

The decision-making matrix for ten criteria with weights is given in Table 7, while the decision-making matrix for eight criteria with weights is given in Table 8.

**Table 7.** Decision-making matrix case for ten criteria used.

| Criterion | $f_1$ | $f_2$ | $f_3$ | $f_4$ | $f_5$ | $f_6$ | $f_7$ | $f_8$ | $f_9$ | $f_{10}$ |
|---|---|---|---|---|---|---|---|---|---|---|
| Unit | % | ha/parcel | parcel | ha | % | % | % | EUR/ha | Year | Un. N |
| Weight | 0.184 | 0.184 | 0.184 | 0.038 | 0.065 | 0.038 | 0.065 | 0.112 | 0.065 | 0.065 |
| Goal | max | min | max | max | max | max | max | min | min | max |
| Alternative | | | | | | | | | | |
| 800317 | 25.54 | 0.49 | 10.12 | 4.93 | 10.99 | 53.53 | 74.45 | 142.0 | 27.0 | 0.37 |
| 800660 | 75.82 | 0.56 | 6.09 | 3.39 | 16.42 | 21.98 | 75.23 | 159.1 | 10.2 | 0.98 |
| 800333 | 74.39 | 1.30 | 3.22 | 4.19 | 2.44 | 29.66 | 5.65 | 133.5 | 8.7 | 1.15 |
| 800643 | 60.71 | 0.43 | 5.47 | 2.35 | 9.71 | 25.18 | 51.36 | 160.5 | 12.8 | 0.78 |
| 800732 | 77.03 | 0.60 | 3.18 | 1.91 | 5.41 | 38.16 | 51.36 | 228.9 | 14.4 | 0.69 |
| 800449 | 57.25 | 0.46 | 5.44 | 2.53 | 10.97 | 23.23 | 89.35 | 162.3 | 13.7 | 0.73 |
| 800678 | 84.04 | 0.92 | 7.02 | 6.48 | 16.35 | 48.00 | 89.17 | 132.1 | 7.6 | 1.31 |
| 800392 | 53.70 | 0.41 | 7.21 | 2.94 | 13.89 | 24.08 | 81.17 | 171.6 | 15.5 | 0.65 |
| 800465 | 79.43 | 0.43 | 5.51 | 2.39 | 10.88 | 15.69 | 36.75 | 133.3 | 8.1 | 1.23 |
| 800473 | 11.76 | 0.80 | 4.44 | 3.57 | 2.88 | 75.24 | 89.14 | 211.4 | 87.1 | 0.11 |
| 800481 | 68.56 | 0.47 | 13.86 | 6.24 | 24.52 | 23.89 | 80.56 | 173.1 | 12.2 | 0.82 |
| 800503 | 87.27 | 0.36 | 9.43 | 3.43 | 13.48 | 9.54 | 55.34 | 133.4 | 7.4 | 1.35 |
| 800708 | 76.39 | 0.32 | 6.17 | 2.01 | 8.71 | 16.05 | 45.02 | 139.9 | 9.1 | 1.10 |
| 800724 | 64.79 | 0.42 | 17.17 | 7.28 | 5.13 | 59.86 | 45.02 | 204.9 | 15.3 | 0.65 |
| 800341 | 78.94 | 0.50 | 5.61 | 2.78 | 8.23 | 19.22 | 28.45 | 141.1 | 8.7 | 1.15 |

**Table 8.** Decision-making matrix case for eight criteria used.

| Criterion | $f_1$ | $f_2$ | $f_3$ | $f_4$ | $f_5$ | $f_6$ | $f_7$ | $f_8$ |
|---|---|---|---|---|---|---|---|---|
| Unit | % | ha/parc | parc | ha | % | % | % | EUR/ha |
| Weight | 0.210 | 0.210 | 0.210 | 0.046 | 0.076 | 0.046 | 0.076 | 0.126 |
| Goal | max | min | max | max | max | max | max | min |
| Alternative | | | | | | | | |
| 800317 | 25.54 | 0.49 | 10.12 | 4.93 | 10.99 | 53.53 | 74.45 | 142.0 |
| 800660 | 75.82 | 0.56 | 6.09 | 3.39 | 16.42 | 21.98 | 75.23 | 159.1 |
| 800333 | 74.39 | 1.30 | 3.22 | 4.19 | 2.44 | 29.66 | 5.65 | 133.5 |
| 800643 | 60.71 | 0.43 | 5.47 | 2.35 | 9.71 | 25.18 | 51.36 | 160.5 |
| 800732 | 77.03 | 0.60 | 3.18 | 1.91 | 5.41 | 38.16 | 51.36 | 228.9 |
| 800449 | 57.25 | 0.46 | 5.44 | 2.53 | 10.97 | 23.23 | 89.35 | 162.3 |
| 800678 | 84.04 | 0.92 | 7.02 | 6.48 | 16.35 | 48.00 | 89.17 | 132.1 |
| 800392 | 53.70 | 0.41 | 7.21 | 2.94 | 13.89 | 24.08 | 81.17 | 171.6 |
| 800465 | 79.43 | 0.43 | 5.51 | 2.39 | 10.88 | 15.69 | 36.75 | 133.3 |
| 800473 | 11.76 | 0.80 | 4.44 | 3.57 | 2.88 | 75.24 | 89.14 | 211.4 |
| 800481 | 68.56 | 0.47 | 13.86 | 6.24 | 24.52 | 23.89 | 80.56 | 173.1 |
| 800503 | 87.27 | 0.36 | 9.43 | 3.43 | 13.48 | 9.54 | 55.34 | 133.4 |
| 800708 | 76.39 | 0.32 | 6.17 | 2.01 | 8.71 | 16.05 | 45.02 | 139.9 |
| 800724 | 64.79 | 0.42 | 17.17 | 7.28 | 5.13 | 59.86 | 45.02 | 204.9 |
| 800341 | 78.94 | 0.50 | 5.61 | 2.78 | 8.23 | 19.22 | 28.45 | 141.1 |

Utilizing model explained above, the ranking results for 15 cadastral municipalities ranked by six MCA analysis models are given in the following tables. Table 9 shows the rankings obtained by 10 criteria, while Table 10 shows the ranking obtained by 8 criteria.

**Table 9.** Ranking lists and standard deviations of alternative ranks with ten criteria.

| CM_ID \ Method | AHP | VIKOR | COPRAS | ELECTRE | TOPSIS | SAW | $\sigma i$ |
|---|---|---|---|---|---|---|---|
| 800317 | 5 | 11 | 11 | 8 | 8 | 12 | 2.64 |
| 800660 | 7 | 7 | 7 | 10 | 7 | 7 | 1.22 |
| 800333 | 14 | 14 | 14 | 13 | 14 | 14 | 0.41 |
| 800643 | 12 | 12 | 12 | 12 | 10 | 11 | 0.84 |
| 800732 | 13 | 13 | 13 | 14 | 13 | 13 | 0.41 |
| 800449 | 11 | 10 | 10 | 11 | 12 | 10 | 0.82 |
| 800678 | 4 | 4 | 4 | 2 | 11 | 3 | 3.20 |
| 800392 | 8 | 9 | 8 | 6 | 5 | 9 | 1.64 |
| 800465 | 6 | 6 | 6 | 5 | 6 | 6 | 0.41 |
| 800473 | 15 | 15 | 15 | 15 | 15 | 15 | 0.00 |
| 800481 | 1 | 1 | 1 | 3 | 1 | 2 | 0.84 |
| 800503 | 3 | 2 | 2 | 1 | 3 | 1 | 0.89 |
| 800708 | 10 | 5 | 5 | 4 | 4 | 5 | 2.26 |
| 800724 | 2 | 3 | 3 | 7 | 2 | 4 | 1.87 |
| 800341 | 9 | 8 | 9 | 9 | 9 | 8 | 0.52 |
| | | | | | | $\sigma_{max}$ | 3.20 |
| | | | | | | $\bar{\sigma}$ | 1.20 |

Figures 1 and 2 show the position of cadastral municipalities with their best and worst rank, obtained by formula (5). The asymmetry in ranks around the average values is caused by the fact that CM could not be ranked better than the first position and worse than the last position (in this case, 15). In this case study, it is obtained that in the case of ten criteria, only two CM are worst ranking smaller than the limit line (CM with identifiers 800481 and 800503), while in the case of eight criteria, three CM are worst ranking smaller the limit line (CM with identifiers 800481, 800503 and 800724). In this proposed model of decision making, those CMs could be considered reliably determined candidates for land

consolidation. For the other three places left, it is possible to rank CM candidates by the following criteria:

- Best "worse" ranking;
- Best "best" ranking;
- Best "average" ranking;
- Utilizing the combination of these criteria.

**Table 10.** Ranking lists and standard deviations of alternative ranks with eight criteria.

| Method<br>CM_ID | AHP | VIKOR | COPRAS | ELECTRE | TOPSIS | SAW | σi |
|---|---|---|---|---|---|---|---|
| 800317 | 5 | 8 | 6 | 6 | 5 | 9 | 1.64 |
| 800660 | 7 | 6 | 7 | 10 | 8 | 6 | 1.51 |
| 800333 | 14 | 14 | 15 | 15 | 15 | 14 | 0.55 |
| 800643 | 12 | 12 | 12 | 11 | 10 | 12 | 0.84 |
| 800732 | 13 | 13 | 13 | 13 | 13 | 13 | 0.00 |
| 800449 | 11 | 11 | 10 | 12 | 11 | 10 | 0.75 |
| 800678 | 4 | 4 | 4 | 2 | 12 | 4 | 3.52 |
| 800392 | 6 | 9 | 8 | 5 | 6 | 8 | 1.55 |
| 800465 | 8 | 7 | 9 | 8 | 7 | 7 | 0.82 |
| 800473 | 15 | 15 | 14 | 14 | 14 | 15 | 0.55 |
| 800481 | 2 | 1 | 1 | 3 | 1 | 1 | 0.84 |
| 800503 | 3 | 2 | 3 | 1 | 3 | 2 | 0.82 |
| 800708 | 10 | 5 | 5 | 7 | 4 | 5 | 2.19 |
| 800724 | 1 | 3 | 2 | 4 | 2 | 3 | 1.05 |
| 800341 | 9 | 10 | 11 | 9 | 9 | 11 | 0.98 |
| | | | | | | σ max | 3.52 |
| | | | | | | $\bar{\sigma}$ | 1.17 |

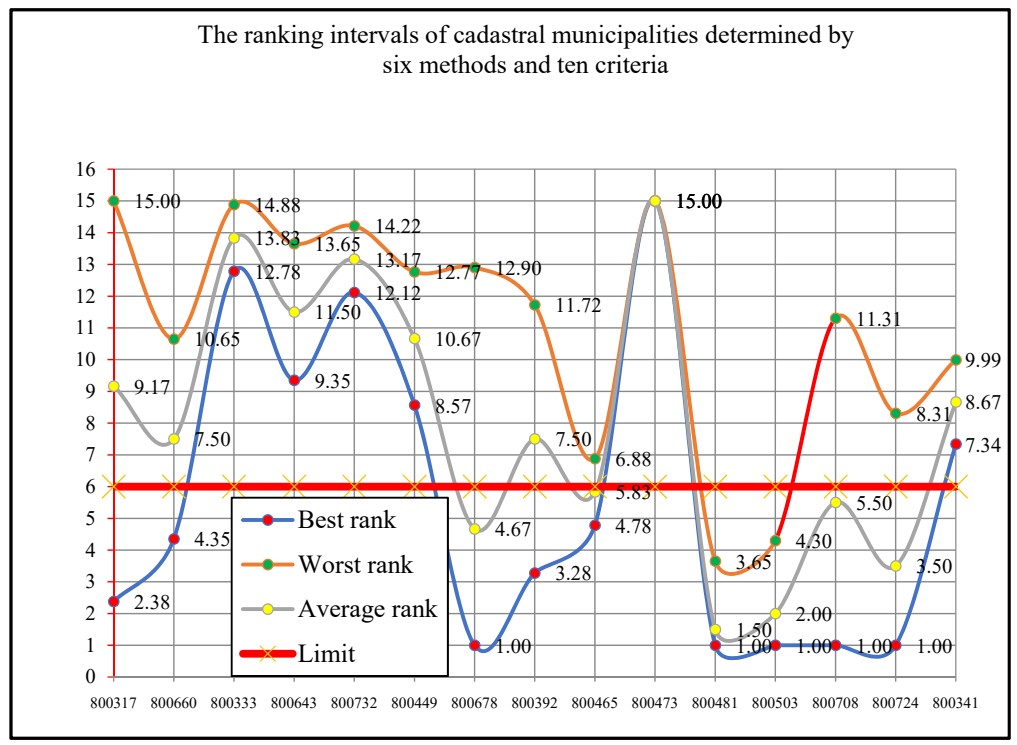

**Figure 1.** The ranking interval of cadastral municipalities with ten criteria.

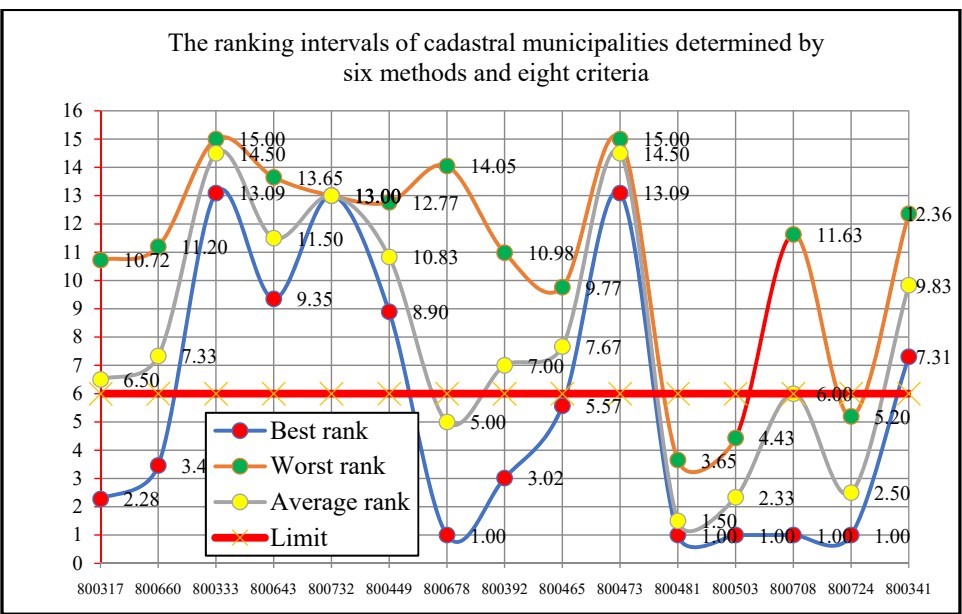

**Figure 2.** The ranking interval of cadastral municipalities with eight criteria.

## 4. Discussion

In this research, the multi-criteria analysis approach was utilized in order to determine the rank of prioritization of cadastral municipalities for realizing land consolidation projects. The basic assumption for providing the analysis is that the resources for land consolidations are limited, and consequently, an objective approach is required. However, the imperfections of multi-criteria analysis methods take a real risk that the wrong decisions could be taken. To avoid this possibility or to reduce risk as much as possible in this research, the idea of random errors in ranking results is introduced. Every utilized method is considered a random process, and consequently, the results of ranking by each utilized method are treated as a result burdened by random error. This allowed the statistical analysis, i.e., by defining the interval in which the rank of each cadastral municipality belongs with equal probability. In our study, the probability is adopted at the level of 95% and of 5 degrees of freedom. Those assumptions resulted in the different positions of priority as well as the different possible intervals where each cadastral municipality belongs. To test the sensitivity of the method, the analysis was provided by calculating ranks with ten and eight criteria.

In the case that financial resources were limited to the six cadastral municipalities that could be land consolidated, it is obvious that six cadastral municipalities could be eliminated in both ten and eight criteria cases; four cadastral municipalities can be adopted with probability of 95%, and five municipalities are candidates for further analysis for final decision making. Final decision making could be provided by using additional criteria or by using obtained statistical parameters (lowest average rank, best rank within intervals, or smallest standard deviation, etc.).

The proposed method reduces the subjective or arbitrary decisions in the process of resource distribution, respecting the imperfections of multi-criteria analysis methods.

## 5. Conclusions

The proposed model for land consolidation projects ranking is based on the assumptions that the financial resources for providing land consolidation are limited and that the multi-criteria analysis methods are not perfect. Imperfections of multi-criteria analysis methods are explicated by the assumption that resulting rankings are burdened by random errors and consequently allow a statistical approach. Additionally, the sensitivity of results was tested by utilizing ten and eight criteria. The results showed that, in this case, final decision should not be changed significantly nevertheless if ten or eight criteria were

performed in the model. After reliable determination of reliable candidates for LC, it is possible that limitation is not fulfilled and that resources for other LC projects still exist. In that case, it is possible to choose CM for LC on the base of best "worst ranking", best "best ranking", best "average ranking" or to combine these criteria for the final decision. This model of decision making provides the base for objective decision making and significantly reduces the need for arbitrary decision making in the process of CM ranking.

**Author Contributions:** Conceptualization, G.M. and Z.I.; methodology, G.M. and Z.I.; validation, G.M., Z.I., M.T., J.T. and M.B.; formal analysis, J.T. and M.B.; investigation, G.M., Z.I. and M.T.; resources, G.M. and Z.I.; writing—original draft preparation, G.M. and Z.I.; writing—review and editing, M.T., J.T. and M.B.; visualization, J.T.; supervision, M.T.; project administration, J.T. and M.B.; funding acquisition, G.M., Z.I., M.T., J.T. and M.B. All authors have read and agreed to the published version of the manuscript.

**Funding:** This research received no external funding.

**Institutional Review Board Statement:** Not applicable.

**Informed Consent Statement:** Not applicable.

**Data Availability Statement:** Not applicable.

**Conflicts of Interest:** The authors declare no conflict of interest.

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
