# Peer review of "Optimization Methods as a Base for Decision Making in Land Consolidation Projects Ranking"

_land, doi:10.3390/land11091466_

Round 1
Reviewer 1 Report
In this paper, the authors presented the possibility of choosing a cadastral municipalities (CM) for the land consolidation (LC) process using multi-criteria analysis (MCA). This approach can help decision makers when choosing a CM for which the LC procedure will be implemented, but in the MCA procedure, the criteria (and their importance - weight) on the basis of which the analysis will be performed must be carefully selected, because a different choice of criteria can significantly change analysis results.
In their paper, the authors state that they compared the results taking into account 10 and 8 criteria, respectively. However, 6 criteria are clearly marked in the paper (Table 1 and Table 2) and one more criterion can be guessed (Table 3). The remaining 3 criteria were not explained in the submitted paper. Also, given that the results for 8 or 10 criteria are given, it is not clear whether the analysis with 8 criteria implies that only 2 criteria were rejected from the analysis with 10 criteria, or were completely different criteria or some third possibility used? If the authors used 8 out of 10 criteria, that is, if they rejected 2 criteria from the first analysis, which criteria were rejected and why exactly those criteria?
I believe that the authors should clarify this in their paper, before its publication.
In the attached pdf file with the manuscript of the paper, I have included comments for things that need to be fixed before publication.

Reviewer 2 Report
The article concerns the important issue of methods reducing the subjective decisions in the process of resources distribution respecting the imperfections of multi-criteria analysis methods in land consolidation. Below are the comments to the article:
1. The aim of the article should be clearly defined in the abstract and in the introduction. 2. The literature review should be broadened and the conclusions developed. 3. The description of the research method lacks information on the remaining four criteria out of ten analyzed (f7-f10). There is no explanation as to why 10 and 8 criteria were included in the analysis (why these 8 out of 10).4. In the abstract it is said: ‘In the case study the thirteen cadastral municipalities were researched’. In fact, 15 municipalities were researched.
5. Line 204: students’ distribution - instead of it, it should be rather ‘Student’s distribution’.
6. What does CM mean in table 1? (cadastral municipalities?)
7. Line 69: repetition in the sentence: Concluding that “the ranking results ranking results was more dependent on indicator weights than the choice of MCA”. Also, what is MCA (the abbreviation is not previously explained). 8. There are some linguistic errors, e.g. line 39 ‘According to Yin et al. [3] research based on the 92 peer-reviewed articles’, it should be: ‘According to research by Yin et al. [3] based on the 92 peer-reviewed articles; line 44 ‘In paper Ying et al. [5] stated that’ instead of ‘In their paper …’ or ‘In the paper by Ying et al. ..’, line 62 ‘if there exist the optimization method’ instead of „if there exists..”. 9. Lines 130-132: ‘According to the same literature: “Hierarchical decomposition of complex systems appears to a basic device used by the human mind to cope with diversity.“ ‘.What the same literature? The reference is needed here.Author Response
Please see the attachment

Reviewer 3 Report
In this paper, the authors presented the possibility of choosing a cadastral municipalities (CM) for the land consolidation (LC) process using multi-criteria analysis (MCA). This approach can help decision makers when choosing a CM for which the LC procedure will be implemented, but in the MCA procedure, the criteria (and their importance - weight) on the basis of which the analysis will be performed must be carefully selected, because a different choice of criteria can significantly change analysis results.
In their paper, the authors state that they compared the results taking into account 10 and 8 criteria, respectively. However, 6 criteria are clearly marked in the paper (Table 1 and Table 2) and one more criterion can be guessed (Table 3). The remaining 3 criteria were not explained in the submitted paper. Also, given that the results for 8 or 10 criteria are given, it is not clear whether the analysis with 8 criteria implies that only 2 criteria were rejected from the analysis with 10 criteria, or were completely different criteria or some third possibility used? If the authors used 8 out of 10 criteria, that is, if they rejected 2 criteria from the first analysis, which criteria were rejected and why exactly those criteria?
I believe that the authors should clarify this in their paper, before its publication.
In the attached pdf file with the manuscript of the paper, I have included comments for things that need to be fixed before publication.

Reviewer 4 Report
Add the study's objective more clearly in the abstract and in the introduction.
Also add paragraph for explanation of graphics in the Result part,
Round 2
Reviewer 1 Report
I am satisfied with the corrections made by the authors.